# Severe fluctuation in mean perfusion pressure is associated with increased risk of in-hospital mortality in critically ill patients with central venous pressure monitoring: A retrospective observational study

Yudie Peng[☯], Buyun Wu[☯], Changying Xing, Huijuan Mao[ID]*

Department of Nephrology, The First Affiliated Hospital of Nanjing Medical University, Jiangsu Province Hospital, Nanjing, China

☯ These authors contributed equally to this work.
* maohuijuan72@hotmail.com, huijuanmao@126.com

## Abstract

**Data Availability Statement:** The datasets generated and analyzed during the current study

### Background

The mean perfusion pressure (MPP) was recently proposed to personalize tissue perfusion pressure management in critically ill patients. Severe fluctuation in MPP may be associated with adverse outcomes. We sought to determine if higher MPP variability was correlated with increased mortality in critically ill patients with CVP monitoring.

### Methods

We designed a retrospective observational study and analyzed data stored in the eICU Collaborative Research Database. Validation test was conducted in MIMIC-III database. The exposure was the coefficient of variation (CV) of MPP in the primary analyses, using the first 24 hours MPP data recorded within 72 hours in the first ICU stay. Primary endpoint was in-hospital mortality.

### Results

A total of 6,111 patients were included. The in-hospital mortality of 17.6% and the median MPP-CV was 12.3%. Non-survivors had significantly higher MPP-CV than survivors (13.0% vs 12.2%, p<0.001). After accounting for confounders, the highest MPP-CV in decile (CV > 19.2%) were associated with increased risk of hospital mortality compared with those in the fifth and sixth decile (adjusted OR: 1.38, 95% CI: 1.07–1.78). These relationships remained remarkable in the multiple sensitivity analyses. The validation test with 4,153 individuals also confirmed the results when MPP-CV > 21.3% (adjusted OR: 1.46, 95% CI: 1.05–2.03).

are available in the eICU-CRD repository, DOI: 10.1038/sdata.2018.178.

**Funding:** The present study was supported by the Priority Academic Program Development (PAPD) of Jiangsu Higher Education Institutions (CN), General Project of the National Natural Science Foundation of China (81970639, 82151320), received by Huijuan Mao. The funders had no role in study design, data collection and analysis, decision to publish, or preparation of the manuscript.

**Competing interests:** The authors have declared that no competing interests exist.

**Abbreviations:** AKI, acute kidney injury; BMI, body mass index; BP, blood pressure; BPV, blood pressure variability; CI, confidence interval; CV, coefficient of variation; CVP, central venous pressure; ICU, intensive care unit; MAP, mean arterial pressure; MPP, mean perfusion pressure; MPPV, mean perfusion pressure variability; OR, odds ratio; SBP, systolic blood pressure; SOFA, Sequential Organ Failure Assessment; TWA-MPP, time-weighted average mean perfusion pressure; VIM, variation independent of the mean.

## Conclusions

Severe fluctuation in MPP was associated with increased short-term mortality in critically ill patients with CVP monitoring.

## Background

Adequate organ perfusion is essential for human, with mean arterial pressure (MAP) as a substitute index for blood perfusion of terminal organs [1]. However, MAP has some physiological deficiencies, especially the failure to consider venous outflow pressure. Central venous pressure (CVP), an indicator of the outflow pressure, remains the most commonly used variable to monitor fluid status and guide fluid resuscitation for critically ill patients [2], especially for those who experience severe trauma, shock, acute circulatory failure, all kinds of major operations and rapid fluid resuscitation [3]. Obtained by calculating the difference between MAP and CVP, mean perfusion pressure (MPP) was recently proposed to personalize tissue perfusion pressure management instead of MAP in critically ill patients [4, 5]. The formula can be expressed as MPP = MAP–CVP. As a novel marker for perfusion pressure, lower MPP was associated with acute kidney injury [6–9], but more knowledge about MPP is warranted.

As an inherent physiological property, the fluctuation of MPP existed and is theoretically associated with outcomes of critically ill patients. In effect, MPP is determined approximately by the product of cardiac output (CO) and systemic vascular resistance (SVR) [7, 10, 11]. Any circulatory condition that affects either of the two factors also affects the fluctuation of MPP. Therefore, a severe fluctuation in MPP, which represents severe hemodynamic instability, may be associated with an adverse prognosis such as deterioration of renal function [12]. Up to now, the threshold at which MPP variability (MPPV) to be clinically significant and the population who are more susceptible to abnormal variability remains unclear.

Therefore, we sought to describe the distribution of MPPV among critically ill patients with CVP monitoring and explore the relationship between MPPV and hospital mortality. We hypothesized that higher MPPV was correlated with increased risk of short-term mortality in these patients.

## Methods

### Study population

This study utilized data stored in the eICU Collaborative Research Database (eICU-CRD) v2.0 [13], a unique and publicly accessible multicenter database covering more than 200,000 ICU admissions. The data stored in the database was collected through the Philips eICU program, a critical care telehealth program that delivers information to caregivers at the bedside. Vital signs were generally interfaced as 1-minute averages, and archived into the database as 5-minute median values [14]. The inclusion criteria were (1) age 16 years or more; (2) at least 24 hours of continuous MAP and CVP invasive monitoring within the first 72 hours in the first ICU stay and (3) at least 20 MPP readings in the daytime and at least seven in the nighttime [15]. Daytime is defined as 7 am to 11 pm, otherwise as nighttime. Those who received dialysis, died during the first 24 hours, were complicated with chronic kidney disease stage 5, intracranial hypertension, abdominal compartment syndrome and with incomplete data or extreme MPP data were excluded. Extreme MPP refers to the values of MAP not between 0 mmHg to 150 mmHg, and the values of CVP not between -10 mmHg to 50 mmHg. Patients with CKD

stage 5 were excluded because they may undergo dialysis, which will significantly affect MPP and increase variability.

## Data extraction

We extracted MPP data, demographic data, baseline ICU characteristics, Charlson comorbidity index [16], and admission illness severity score (the Sequential Organ Failure Assessment (SOFA) [17]). Criteria for sepsis were defined based on those described earlier by Angus et al [18] instead of sepsis 3.0 because most microbiology data was unavailable in eICU-CRD. Additionally, the need for mechanical ventilation, the incidence of AKI, use of vasopressor, antihypertensive drugs, and sedatives were also collected. As MPP is a dynamic process, time-weighted average MPP (TWA-MPP) during the first 24 hours of ICU stay was calculated as the area under the MPP–versus–time plot as follows to truly reflect the average level of MPP.

$$TWA = [(t_2-t_1)(X_1 + X_2)/2 + (t_3-t_2)(X_2 + X_3)/2 + \ldots + (t_n-t_{n-1})(X_{n-1} + X_n)/2]/(t_n-t_1)$$

where $X_n$ is the value of the variable of interest at the timepoint $t_n$.

## Data cleaning

We chose the values of MAP between 0 mmHg and 150 mmHg, and the values of CVP between -10 mmHg and 50 mmHg.

## Exposure

Short-term MPPV was measured as the coefficient of variation (CV) of 24-hour MPP data (MPP-CV), defined as the standard deviation (SD) divided by the mean MPP value.

## Outcomes

The primary outcome was in-hospital mortality.

## Statistical analysis

This is a post hoc analysis. Statistical analyses were performed using R version 3.63 (R Foundation for Statistical Computing, Vienna, Austria; www.r-project.org). Firstly, the baseline characteristics were compared between survivors and non-survivors. Categorical variables were presented as percentages and compared using a chi-square test. Continuous variables were expressed as median (25th, 75th percentile) and compared using Wilcoxon rank-sum test. To get a better understanding of the relationship between MPPV and TWA-MPP as well as other blood pressure variability (BPV), we used correlation matrices to show the correlation coefficient and then analyzed the association between MAP variability (MAPV) and prognosis.

Secondly, generalized additive models with a logit link function were built to plot associations between MPP-CV and in-hospital mortality, adjusted by age, gender, BMI, ethnicity, Charlson comorbidity index, SOFA score, admission type (elective surgery, emergency surgery or medicine), cardiovascular surgery, history of tachyarrhythmia, sepsis, incidence of AKI in the first day of ICU admission, the need for mechanical ventilation, the use of vasopressor, antihypertensive drug, sedatives, and TWA-MPP.

Thirdly, taking MPP-CV as classification variables, we used multivariable logistic regression models to assess the relationship between the hospital mortality and deciles of each parameter in which the median two deciles, the fifth decile together with the sixth decile, were chosen as reference. The multivariable logistic regression models were adjusted by the same variables mentioned above.

## Subgroup and sensitivity analyses

Subgroup analyses of increased MPP-CV were conducted in patients who were male or female, elderly (age $\geq$ 65 years) or not, with or without hypertension, sepsis, higher than median SOFA score or not on the first day of ICU admission, admission type (surgical or medical), cardiovascular surgery or not.

Variation independent of the mean (VIM) [19] of 24-hour MPP data (MPP-VIM) was also analyzed in the sensitivity analyses. Both of the two indicators (CV and VIM) are considered to be relatively independent of the mean value [19]. Detailed formulas are displayed in S1 Table. Furthermore, as circadian rhythm exists in blood pressure, the association between daytime or nighttime MPP-CV and hospital mortality were also analyzed to observe whether the association between MPPV and prognosis is solely contributed by daytime or nighttime MPPV. Finally, based on the median MPP-CV (12%) during the first 12 hours of the 24 hours, we categorized the patients into initial low variability (MPP-CV < 12% in the first 12 hours of the 24 hours) groups and initial high variability (MPP-CV > 12% in the first 12 hours of the 24 hours) groups. Based on the MPP-CV in the second 12 hours of the 24 hours, the initial low variability group was further categorized into persistently low (MPP-CV < 12% in the second 12 hours of the 24 hours) group (group 1) and increasing (MPP-CV > 12% in the second 12 hours of the 24 hours) group (group 2); the initial high variability group was further categorized into decreasing (MPP-CV < 12% in the second 12 hours of the 24 hours) group (group 3) and persistently high (MPP-CV > 12% in the second 12 hours of the 24 hours) group (group 4). By grouping, we tried to observe the difference of hospital mortality under different change modes.

There were missing values for body mass index (BMI) (3.2%) and multiple imputation was used to handle the missing values with the mice package in R. For all analyses, a two-tailed P value less than 0.05 was considered statistically significant.

## Validation test

The Medical Information Mart for Intensive Care (MIMIC)-III [20, 21] database (version v1.4) covers 46,476 patients out of 61,532 ICU admissions from 2001 to 2012 at the Beth Israel Deaconess Medical Center in Boston, MA, USA. Vital signs measurements were made at the bedside about one data point per hour. Based on the same inclusion and exclusion criteria, we conducted the validation test on patients from MIMIC-III.

## Ethics approval and consent to participate

The study was conducted entirely on the publicly available, third-party anonymous public databases. The ethics committee of our hospital waived the requirement for approval of this study (2021-QT-08). To apply for access to the database, we completed the National Institutes of Health's web-based course and passed the Protecting Human Research Participants exam (record ID. 32559175, ID. 38120064). All methods were performed in accordance with the relevant guidelines and regulations.

## Results

### Patient characteristics

After reviewing 166,355 first ICU stays in eICU-CRD, we finally included 6,111 fulfilling the inclusion and exclusion criteria (Fig 1). The baseline characteristics between survivors and non-survivors are shown in Table 1. The survivors were, on average, younger, predominantly male and higher in BMI. Non-survivors were significantly complicated with more

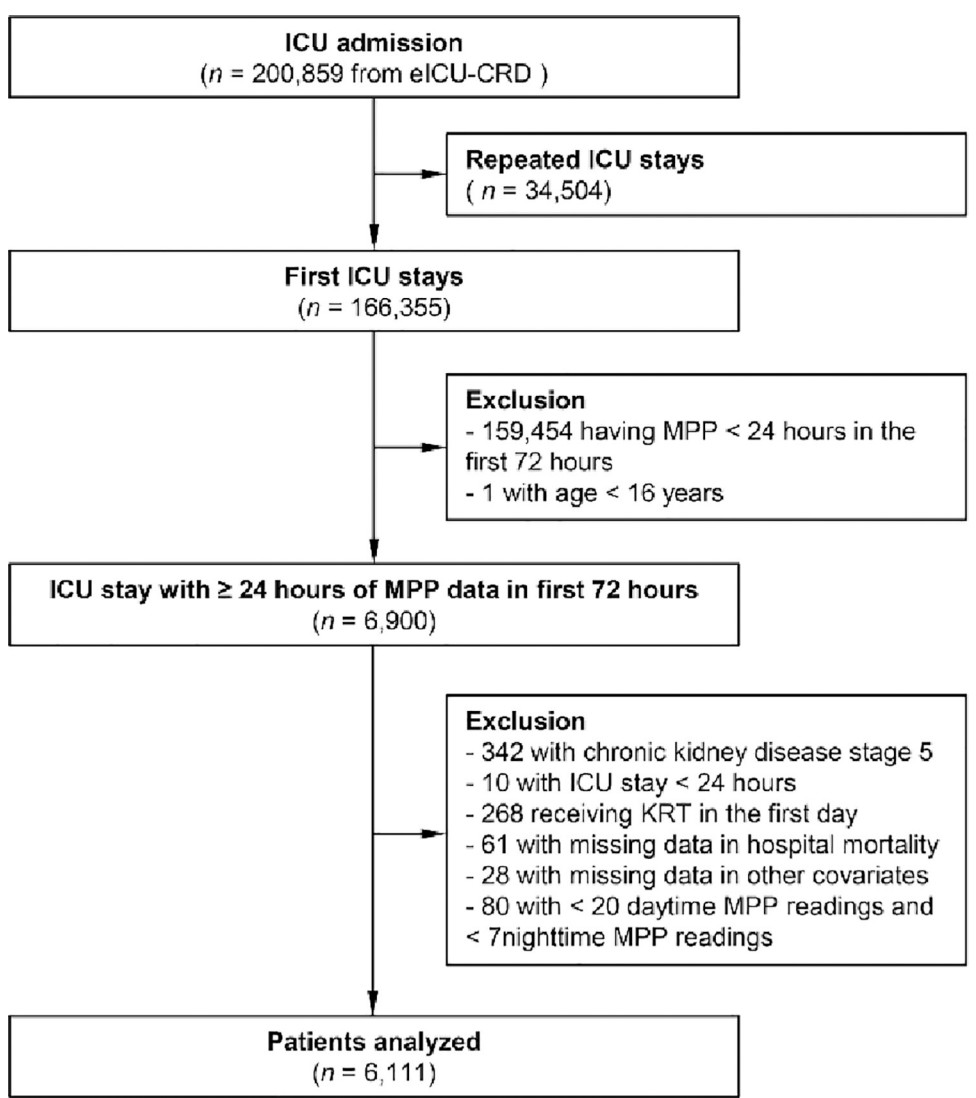

**Fig 1. Patient flow chart.**

comorbidities, more severe in SOFA score, needing more support (mechanical ventilation and vasopressor), less antihypertensive drug use, higher incidence of AKI and sepsis, and lower MPP compared with survivors. More non-survivors had a tachyarrhythmia history and less use of sedatives, but the difference did not reach statistical significance. Other information about hospitals, initial diagnosis, comorbidities, and MPP data of the whole cohort was listed in S2 Table.

The median of the MPP-CV was 12.3% in the whole cohort. The 10th and 90th percentile for MPP-CV were 8.1% and 19.2%, respectively. The non-survivors had higher MPP-CV (13.0% vs 12.2%, p<0.001) as compared with survivors.

## Association with TWA-MPP and other BPV

The correlation matrix showed us that the correlation coefficient between MPP-CV and MPP-VIM was 0.98, which was very strong. There was no correlation between the two MPPV parameters and the TWA-MPP (S1 Fig). We also explored the correlation coefficients between

**Table 1. Baseline characteristics of the study population in the first 24 hours and MPP characteristics of the exposure time among survivors and non-survivors.**

| Variables | Survivors | Non-Survivors | p |
|---|---|---|---|
| N | 5034 | 1077 | |
| Age | 66 (56, 75) | 68 (57, 77) | **<0.001** |
| Male (%) | 3064 (60.9) | 615 (57.1) | **0.024** |
| BMI (kg/m²) | 28.6 (24.6, 33.5) | 28.2 (23.6, 33.6) | **0.019** |
| White (%) | 3809 (75.7) | 851 (79.0) | **0.021** |
| Charlson | 1 (0, 2) | 1 (0, 3) | **0.006** |
| SOFA score | 8 (6, 10) | 10 (8, 13) | **<0.001** |
| Admission type | | | **<0.001** |
| Medicine | 2190 (43.5) | 821 (76.2) | |
| Elective surgery | 2562 (50.9) | 200 (18.6) | |
| Urgent surgery | 282 (5.6) | 56 (5.2) | |
| Cardiovascular surgery | 1769 (35.1) | 116 (10.8) | **<0.001** |
| Sepsis (%) | 972 (19.3) | 365 (33.9) | **<0.001** |
| History of Tachyarrhythmia (%) | 674 (13.4) | 169 (15.7) | **0.052** |
| First day AKI (%) | 1851 (36.8) | 593 (55.1) | **<0.001** |
| Ventilation (%) | 3526 (70.0) | 882 (81.9) | **<0.001** |
| Vasopressors (%) | 2016 (40.0) | 549 (51.0) | **<0.001** |
| Sedatives (%) | 2555 (50.8) | 527 (48.9) | **0.293** |
| Antihypertensive drugs (%) | 1726 (34.3) | 255 (23.7) | **<0.001** |
| Measurement times of MPP | 284 (264, 288) | 279 (246, 288) | **<0.001** |
| TWA MPP (mmHg) | 63.3 (57.8, 70.0) | 60.5 (53.7, 69.0) | **<0.001** |
| MPP-CV (%) | 12.2 (9.9, 15.3) | 13.0 (10.3, 16.8) | **<0.001** |
| MPP-VIM (units) | 0.40 (0.32, 0.50) | 0.42 (0.33, 0.54) | **<0.001** |

Continuous variables were expressed as median (interquartile range) as the distributions are skewed and categorical variables were expressed as number (percentage).
AKI: acute kidney injury; BMI: body mass index; CV: coefficient of variation; ICU: intensive care unit; MPP: mean perfusion pressure; SOFA: Sequential Organ Failure Assessment; TWA: time weighted-average; VIM: variation independent of the mean.

MPPV and other BPV. Among them, MAP-CV had the highest correlation coefficient ($r = 0.77$, $r^2 = 0.60$) with MPP-CV. Although CVP is also a part of MPP in calculation, the correlation between CVP-CV and MPP-CV was weak ($r = 0.08$, $r^2 = 0.006$).

## Association with hospital mortality

Before and after adjusting for all the confounders, we found that hospital mortality increased when the MPP-CV increased (Fig 2A). After grouping in deciles (Fig 2B), univariate logistic regression revealed that higher MPP-CV (CV > 19.2%) were related to an increase in the risk of hospital mortality compared with the fifth and sixth decile (adjusted odds ratio [OR] in the tenth decile: 1.91, 95% confidence interval [Cl]:1.51–2.41). Multivariable logistic regression also revealed an increase in the risk of hospital mortality when MPP-CV > 19.2% (adjusted OR in the tenth decile: 1.38, 95% Cl:1.07–1.78).

Considering the high correlation between MPPV and MAPV, we also analyzed the relationship between MAP-CV and prognosis in two databases. In eICU-CRD database, MAP-CV and mortality showed a U-shaped curve as compared to the median two deciles. In the MIMIC database, however, there was no significant correlation between increased MAP-CV and prognosis (S2 Fig). In terms of predicting hospital mortality, MPPV has a slightly advantage than MAPV (S3 Table).

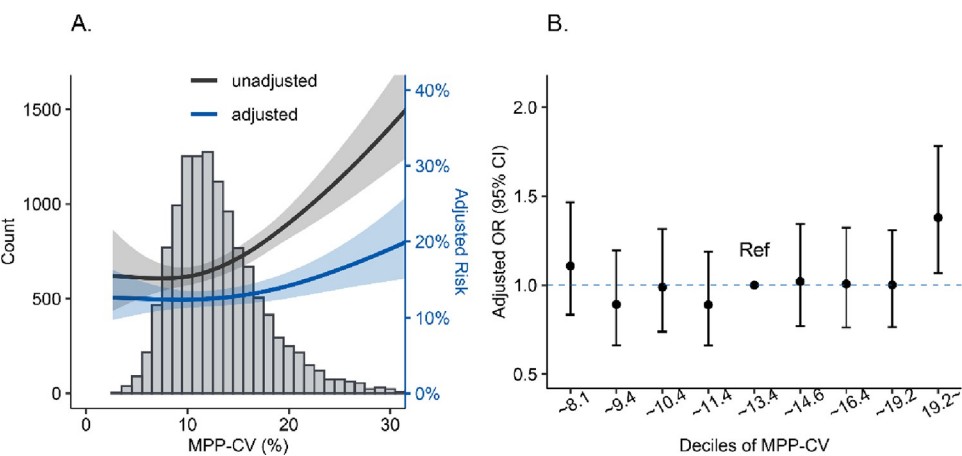

**Fig 2. The association between MPP-CV and in-hospital mortality.** A. The associations between in-hospital mortality risk and MPP-CV fitted by general additive models and the histograms of MPP-CV. B. The logistic regression analyses of the associations between adjusted in-hospital mortality risk and deciles of MPP-CV, taking the median two groups (the fifth decile and the sixth decile) as reference. The above associations were adjusted by age, gender, BMI, ethnicity, Charlson comorbidity index, SOFA score, admission type (elective surgery, emergency surgery or medicine), cardiovascular surgery, history of tachyarrhythmia, sepsis, incidence of AKI in the first day of ICU admission, the need for mechanical ventilation, the use of vasopressor, antihypertensive drug, sedatives and time-weighted average MPP.

## Sensitivity and subgroup analyses

For the sensitivity analyses, we firstly chose MPP-VIM as another variability parameter. The median of the MPP-VIM was 0.40 units in the whole cohort. The non-survivors also had VIM (0.42 units vs 0.40 units, p<0.001) as compared with survivors. We observed a similar trend as MPP-CV in the relationship between MPP-VIM and hospital mortality (S3A Fig). Multivariable logistic regression furtherly confirmed our findings, higher MPP-VIM (VIM > 0.62 units) were related to an increase in the risk of hospital mortality compared with the fifth and sixth decile (adjusted OR in the tenth decile: 1.42, 95% Cl:1.10–1.84 (S3B Fig). Secondly, the results of the association between day and night MPP-CV and hospital mortality still showed good consistency (S4 Fig).

In the subgroup analyses (Fig 3), higher MPP-CV is associated with higher risk of in-hospital mortality in the patients with a SOFA score ≥ 8. In contrast, high MPP-CV did not increase the risk of in-hospital mortality in sepsis patients. The results drawn in MPP-VIM are consistent (S5 Fig).

## MPP-CV change modes

According to the MPP-CV in the two periods of the first 24 hours with MPP data recorded (0–12 hours and 12-24hours), the patients were divided into four groups (Fig 4). Most patients belonged to the persistently low variability group (N = 2302). The decreasing group had the smallest number of patients (N = 791). Patients with persistently high variability had the highest hospital mortality (21.3%), and patients with persistently low variability had the lowest hospital mortality (15.7%).

## Validation test

In the cohort of MIMIC-III database, 4,153 patients were enrolled with the same inclusion and exclusion criteria (Fig 5A). Although the overall MPP-CV value of MIMIC-III cohort is

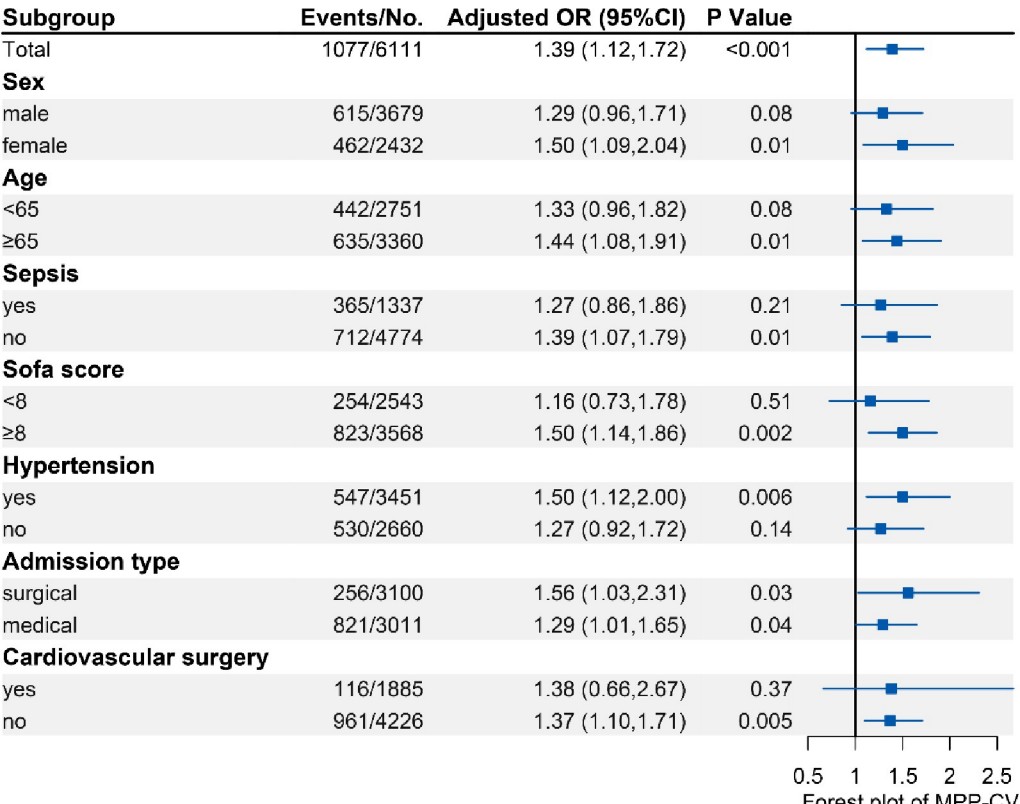

**Fig 3. Adjusted odds ratios and 95% CIs for hospital mortality associated with the increased MPP-CV in different subgroups.** Subgroup analyses of increased MPP-CV were conducted in patients who were male or female, elderly (age ≥ 65 years) or not, with or without hypertension, sepsis, higher than median SOFA score or not on the first day of ICU admission, admission type (surgical or medical), cardiovascular surgery or not. The above associations were adjusted by age, gender, BMI, ethnicity, Charlson comorbidity index, SOFA score, admission type (elective surgery, emergency surgery or medicine), cardiovascular surgery, history of tachyarrhythmia, sepsis, incidence of AKI in the first day of ICU admission, the need for mechanical ventilation, the use of vasopressor, antihypertensive drug, sedatives and time-weighted average MPP.

slightly higher in distribution, it can still be observed that hospital mortality increased when the MPP-CV increased (Fig 5B), and the highest decile of MPP-CV (CV > 21.3%) were related to an increase in the risk of hospital mortality compared with the fifth and sixth decile (adjusted OR in the tenth decile: 1.46, 95% Cl:1.05–2.03) (Fig 5C).

## Discussion

### Main findings

The MPP was recently proposed to personalized management tissue perfusion pressure instead of MAP in critically ill patients. However, we knew little about the relationship between MPPV and mortality. In this multicenter, retrospective cohort study among critically ill patients with CVP monitoring, we aimed to clarify the clinically significant range of MPPV abnormalities for the first time. We found that the median MPP-CV was 13.2% in critically ill patients during the first 24 hours of CVP monitoring. And severely high MPP-CV that reached around 20% or more in two cohorts, occurring in about 10% of the study participants, was associated with the increased risk of in-hospital mortality.

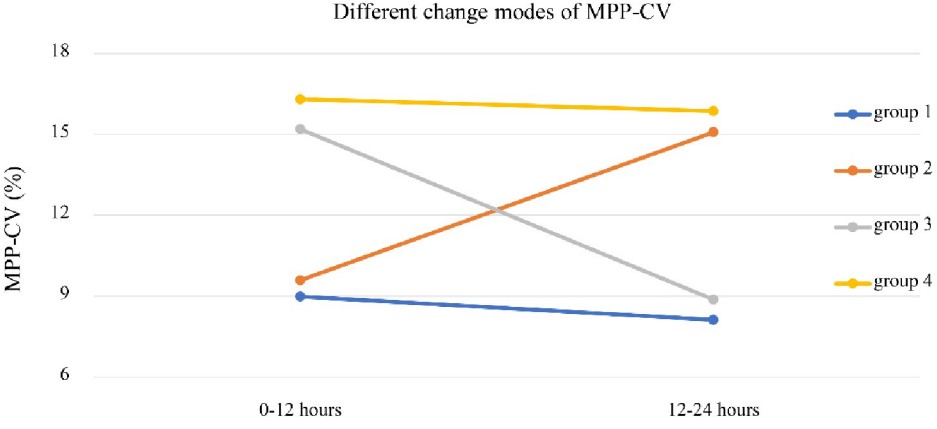

| | group 1 | group 2 | group 3 | group 4 | p value |
|---|---|---|---|---|---|
| Numbers | 2302 | 791 | 1572 | 1438 | |
| 0-12 hours MPP-CV(%) | 8.98 (7.37, 10.42) | 9.59 (8.10, 10.89) | 15.20 (13.43, 17.97) | 16.31 (14.17, 20.30) | <0.001 |
| 12-24 hours MPP-CV(%) | 8.12 (6.43, 8.06) | 15.08 (13.17, 17.96) | 8.87 (7.28, 10.43) | 15.87 (13.63, 19.32) | <0.001 |
| Hospital Mortality (%) | 15.7 | 17.9* | 17.0*‡ | 21.3**‡ | <0.001 |
| ICU Mortality (%) | 11.2 | 13.9 | 12.5 | 16.0*‡ | <0.001 |

*compared to group 1, adjusted $p < 0.05$; † compared to group 2, adjusted $p < 0.05$; ‡ compared to group 3, adjusted $p < 0.05$.

**Fig 4. Different change modes of MPP-CV and the corresponding prognosis.** MPP-CV were expressed as median (interquartile range).

### Implications of study findings

In our study, two variability indicators confirmed the link between high variability and increased risk of hospital mortality. In addition, the same conclusion could also be drawn when analyzing daytime and nighttime MPP-CV separately. The mortality under different variability modes also confirmed the correlation between increased MPPV and prognosis. The exposure was mainly focused on MPP-CV, an indicator of the relative scatter of the values, which is easy to calculate and understand. The risk of short-term mortality increased when MPP-CV (SD/mean) reached around 20% or more. According to our results, if a patient has an average MPP of 60 mmHg and MPP-CV < 20%, then SD should be less than 12 mmHg. That is to say, 95% of the MPP readings should be within the range of 36.5 to 83.5 mmHg (mean ± 1.96 SD) based on a hypothesis of normal distribution of the MPP. Obviously, MPP-CV of over 20% represented a severe fluctuation in MPP, but it indeed occurred in about 10% of the study population in both cohorts. Therefore, it is physiologically acceptable that avoiding severe fluctuation in MPP (MPP-CV < 20%) may be a potential target for better hemodynamic management enhancing the outcomes of these patients.

One previous study has shown that intraoperative systolic BPV was associated with short-term mortality in patients undergoing aortocoronary bypass surgery [22]. Our study mainly focused on the MPP variability in critically ill patients. Why is severe fluctuation in MPP related to adverse outcomes? In effect, MPP is determined approximately by the product of cardiac output and systemic vascular resistance [23]. Any condition of circulation that affects either of these two factors also affects the fluctuation of MPP. The change of blood volume, electrolytes, acute deterioration of cardiac function could affect the cardiac output, and the vasoactive agents, sedatives and pain could impact the SVR. Therefore, severe fluctuation of MPP indirectly represents a significant variation of management in blood volume, cardiac

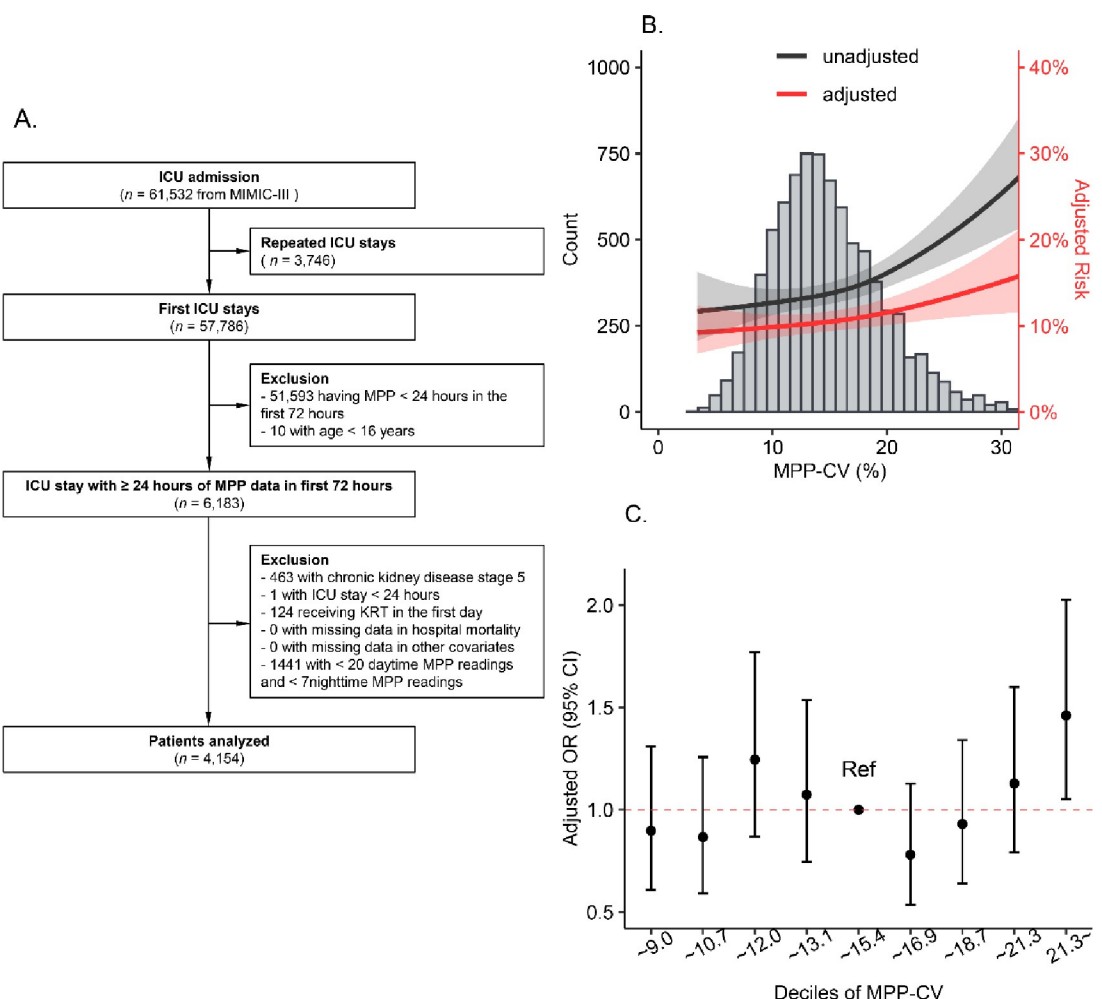

**Fig 5. The validation test of MIMIC-III database.**

function, electrolytes control, use of vasoactive agents, sedatives and pain management, contributing to increased mortality risk. A global perspective pointed out that rapid changes in the infusion rate of vasoactive drugs or clinicians who desired to maintain higher blood pressure levels than expected without proper de-escalation are likely to cause serious adverse complications [24]. The relationship between positive fluid balance and the development or worsening of organ dysfunction as well as excess mortality has also been confirmed [25]. Undoubtedly, severe fluctuation of MPP represents a marker of illness severity to some extent, although the illness severity was accounted for in the multivariable models. Further randomized controlled trials are required to confirm the potential causal relationship between increased MPPV and mortality.

Interestingly, the patients with higher SOFA scores seem to be more susceptible to higher MPPV according to the subgroup analyses. Critically ill patients with SOFA ≥ 8 may be presented with a higher rate of microcirculation dysfunction. In this case, the tissue oxygen extraction capacity is lost, and a more severe or prolonged duration of hypotension will aggravate tissue hypoxia. Therefore, MPP stability management should be strengthened when treating patients with more severe multiple organ dysfunction. Our subgroup analyses also showed that higher MPPV did not associate with in-hospital mortality in patients with sepsis. Our

result disagreed with a previous prospective study, which showed a correlation between early higher SBP complexity and increased risk of 28-day mortality in 51 patients with severe sepsis [26]. But their study only analyzed SBP variability on the first five-minute window. Patients with sepsis are often characterized by increased MPP and MPPV during fluid resuscitation but would not necessarily develop adverse outcomes. Various studies have shown that high short-term to long-term BPV is associated with adverse outcome in patients with hypertension [27–29]. Our study further clarified the relationship between short-term blood pressure variability and poor prognosis in hypertensive population which may be related to arteriosclerosis and decreased ability to regulate blood pressure and made them more vulnerable to ischemia-reperfusion injury.

Most of the variability in MPP can be explained by MAPV. However, the correlation between MAP-CV and prognosis showed no significant difference after multiple adjustments, suggesting that MAPV was of less robustness than MPPV to predict prognosis. Although there is still controversy about fluid resuscitation under the guidance of CVP [30–32], we argue that more focus should be paid to MPPV in critically ill patients with CVP monitoring, as MPP comprehensively reflects the overall perfusion [4].

## Strengths and limitations

This is the first clinical investigation to explore the association between the MPPV and hospital mortality in critically ill patients. The advantage of this post hoc analysis was that both eICU-CRD and MIMIC-III databases contained comprehensive and high-quality data, which guaranteed the reliability of variability calculation. Moreover, the inclusion of the 24-hour measurement ensured that all patients were exposed to a complete diurnal cycle. Finally, we conducted sensitivity analyses and validation test to make the results robust.

Our study has some limitations. First, the post hoc analysis has its inherent defects and unavoidable bias. Second, our study population is limited to the patients with central venous pressure monitoring, who are more severely ill and cannot be extended to the whole population of critically ill patients. Third, it was hard to prove the causal relationship between MPPV and the primary endpoint as the study was observational, despite using two databases to confirm the association. The question of whether MPPV was a marker of severity of illness or a potential target to improve prognosis required randomized trials to answer. Fourth, our study did not account for advanced hemodynamic data such as cardiac index, peripheral vascular resistance and mechanical ventilation parameters like positive end-expiratory pressure.

## Conclusion

Severe MPP fluctuation was associated with short-term mortality in critically ill patients with CVP monitoring. Therefore, it may need to be avoided in the management of critically ill patients.

## Supporting information

**S1 Fig. Correlation matrices of TWA-MPP and MPPV.**
(TIF)

**S2 Fig. The associations between in-hospital mortality risk and MAP-CV in both databases.**
(TIF)

**S3 Fig. The associations between MPP-VIM and in-hospital mortality.**
(TIF)

**S4 Fig. The associations between in-hospital mortality risk and daytime or nighttime MPP-CV.**
(TIF)

**S5 Fig. The associations between in-hospital mortality risk and MPP-VIM in validation test cohort (MIMIC-III database).**
(TIF)

**S6 Fig. Adjusted odds ratios and 95% CIs for hospital mortality associated with the increased MPP-VIM in different subgroups.**
(TIF)

**S1 Table. Calculation formula of variability parameters.** Note: n is the number of MPP readings, $\bar{x}$ is the mean value and w refers to the time of each interval. For VIM, linear regression fitting log (SD) with log (x) was performed. The "k" was the exponential of β0 and the "b" was the β1 of the linear regression model.
(DOCX)

**S2 Table. Other information of the study population.** Continuous variables were expressed as median (interquartile range) as the distributions are skewed and categorical variables were expressed as number (percentage). ICU: intensive care unit; MPP: mean perfusion pressure; TWA: time weighted-average.
(DOCX)

**S3 Table. The comparison of the AUC between MPPV and MAPV in prediction the hospital mortality.** AUC: area under the curve; CI: confidence interval; MPPV: mean perfusion pressure variability; MAPV: mean arterial pressure variability.
(DOCX)

**S1 File. Minimal data set for eICU-CRD.**
(CSV)

**S2 File. Minimal data set for MIMIC-III.**
(CSV)

## Acknowledgments

We thank for the work of researchers at the MIT Laboratory for Computational Physiology, Philips Healthcare, and their collaborators. We also thank Ph.D Jin Liu for his help in statistics.

## Author Contributions

**Conceptualization:** Yudie Peng, Buyun Wu, Huijuan Mao.

**Methodology:** Yudie Peng, Buyun Wu.

**Supervision:** Changying Xing, Huijuan Mao.

**Writing – original draft:** Yudie Peng, Buyun Wu.

**Writing – review & editing:** Huijuan Mao.

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
