## [Decision Letter · Decision Letter 0]

10 Apr 2023

PONE-D-23-05624Severe fluctuation in mean perfusion pressure is associated with increased risk of in-hospital mortality in critically ill patients with central venous pressure monitoring: a retrospective observational studyPLOS ONE

Dear Dr. Mao,

Thank you for submitting your manuscript to PLOS ONE. After careful consideration, we feel that it has merit but does not fully meet PLOS ONE’s publication criteria as it currently stands. Therefore, we invite you to submit a revised version of the manuscript that addresses the points raised during the review process.

We look forward to receiving your revised manuscript.

Kind regards,

Karthik Raghunathan

Academic Editor

PLOS ONE

Journal Requirements:

"The present study was supported by the Priority Academic Program Development (PAPD) of Jiangsu Higher Education Institutions (CN), General Project of the National Natural Science Foundation of China (81970639，82151320), received by Huijuan Mao."

Additional Editor Comments:

The reviewers had mixed reactions to the manuscript but I am prepared to see a revised version. Please be sure to address the reviewer concerns in detail.

Reviewers' comments:

Reviewer's Responses to Questions

**Comments to the Author**

1. Is the manuscript technically sound, and do the data support the conclusions?

Reviewer #1: Yes

Reviewer #2: Yes

Reviewer #3: Partly

2. Has the statistical analysis been performed appropriately and rigorously? 

Reviewer #1: Yes

Reviewer #2: Yes

Reviewer #3: Yes

3. Have the authors made all data underlying the findings in their manuscript fully available?

Reviewer #1: Yes

Reviewer #2: Yes

Reviewer #3: No

4. Is the manuscript presented in an intelligible fashion and written in standard English?

Reviewer #1: Yes

Reviewer #2: Yes

Reviewer #3: No

5. Review Comments to the Author

Reviewer #1: Thank you for the opportunity to review the manuscript, entitled “Severe fluctuation in mean perfusion pressure is associated with increased risk of in-hospital mortality in critically ill patients with central venous pressure monitoring: a retrospective observational study” It is well written and important points are clearly presented. I have the following questions and comments to the authors.

1) It seems that the authors focused on MPP-CV as a primary exposure, but there were many sensitivity analyses that may distract readers. For example, MPP-VIM was equally reported as if it is a primary exposure. All sensitivity and subgroup analyses should be concisely summarized in the sensitivity and subgroup analysis section of the results. Can the authors choose a few important sensitivity and subgroup analyses?

2) Can the authors clarify how MPP was obtained by showing an equation? Some readers may not be familiar with it.

3) The patient characteristics were not clear. The e-ICU data includes heterogenous patients. How many patients were elective admission? How many patients underwent cardiac and non-cardiac surgery? Can the authors adjust for them in the regression analysis?

4) Line 151: Was TWA-MPP adjusted? Why?

5) Line 177: This would not be an external validation. When it comes to external validation, a newly developed prediction score is going to be tested in a different dataset. If the authors can obtain the same information from MIMIC-III, two databases can be combined into one dataset.

6) In this study, looking at ICU mortality would be less important.

7) Line 161: Why daytime and nighttime MPP-CV was measured? Please clarify.

Reviewer #2: The present manuscript is an interesting study in which the authors use a large, multicenter eICU database (with external validation database) to analyze whether the coefficient of variation (CV) of mean perfusion pressure (MPP) (MPP-CV) over 24hrs within first 72hrs of ICU stay correlated with in-hospital mortality. They determined the median MPP-CV for the 2 databases. They also found that patients with high MPP-CV (especially >19%) had higher OR of in hospital mortality. They used a variety of secondary and subgroup analyses to confirm the robustness of their primary finding. They concluded that targeting fewer MPP swings in patients (~20% of average MPP) could be target for therpies. Overall, the topic is important, the manuscript is well-written, data analysis methods overall appear solid, and the findings are significant. Please see comments below for issues that should be addressed.

General comments:

1. Why were patients with CKD5 excluded? No justification is given. The authors should justify exclusions.

2. For exclusions, what does “extreme MPP were excluded” mean? The authors need to specify.

3. What were variables in multivariable model and how were they chosen?

4. Why did the authors pick MPP-CV < or > 12% as a cutoff? Justification is not given in methods though presumably due to median of MPP-CV of 12.3% in whole dataset?

5. Were any analyses pre-specified? This should be indicated in methods.

6. Why did the authors not include hypertension or baseline blood pressure in the model? This seemingly could be a major confounder as patients with hypertension would likely have wider swings in blood pressure. They do account for this in subgroup analysis, but curious why it was not included in the original model? Is this what anti-hypertension drug refers to in the figure legend? The authors should clarify.

7. Did the authors analyze any secondary outcomes? Presumably they have information regarding outcomes such as AKI, mechanical ventilation, etc. These would be meaningful additions.

Reviewer #3: The authors submit a manuscript describing a retrospective analysis of large database of critically ill patients having invasive monitoring. The focus was on mean-perfusion pressure variation (MAP-CVP) and mortality. The found that the average coefficient of variation the first 24 hrs of ICU admission was higher in non-survivors versus survivors. The highest decile of MPP CV was independently associated with mortality.

The topic of this paper is clinically interesting and consistent with other reports. I have a few comments.

1. It would be helpful for the readers to have an understanding of the unique aspects of the question of this study. That is, what new questions the study asks. They mention that there is knowledge that MPP variation is associated with adverse outcomes then how does this paper add to this body of knowledge? Interestingly, there are other data not references showing that fluctuations in BP in patients undergoing surgery is associated with adverse events (I believe the first author was Aronson, S). They did not address MPP to my knowledge.

2. Page 5 in the Methods section line 112: Readers may want to know how you defined extreme MPP data.

3. Page 6, lines 164-165: how did you decide to categorize high and low variability as MPP 12%?

4. An entrance criteria of having both direct blood pressure and CVP present may limit the external validity of the study when those measures are not present. That is why the readers need to know the reason for exclusion from analysis. One wonders what the added value of MPP versus MAP alone might be in the predictive modeling. While I agree with the rationale for MPP is there useful data on MAP CV alone? We do not know if the relationship found in this study is due to reduced perfusion per se versus just MAP or the factors that cause MAP variation. How do we know as an example this is not simply due to HFpEF? Those patients have high variability due to diastolic dysfunction.

5. Lines 292-293. MAP is the product of cardiac output and SVR. MPP is MAP -CVP.

6. Lines 350-351: Making as statement that MPP should be targeted as a means to improve outcome is not really warranted based on these data.

6. PLOS authors have the option to publish the peer review history of their article (what does this mean?). If published, this will include your full peer review and any attached files.

Reviewer #1: No

Reviewer #2: No

Reviewer #3: No

---

## [Author Response · Author response to Decision Letter 0]

17 May 2023

Dear Editor,

 Thank you for inviting us to submit a revised version of our manuscript. We have revised the paper according to the reviewers’ suggestions. The revised version of the paper is attached, with highlighting where changes have been made in the manuscript.

 Below are our specific responses to the reviewers’ comments.

Journal Requirements:

Response: Okay, we ensure that our manuscript meets PLOS ONE's style requirements.

"The present study was supported by the Priority Academic Program Development (PAPD) of Jiangsu Higher Education Institutions (CN), General Project of the National Natural Science Foundation of China (81970639，82151320), received by Huijuan Mao."

Response: Okay, we've added relevant expressions in the cover letter.

Response: Okay. In the revised version, we upload our study’s minimal data set as Supporting Information files. 

Response: Our ethics statement is written in the Methods section (line189-195).

Response: Okay, we include captions for Supporting Information files at the end of your manuscript.

Reviewer #1: Thank you for the opportunity to review the manuscript, entitled “Severe fluctuation in mean perfusion pressure is associated with increased risk of in-hospital mortality in critically ill patients with central venous pressure monitoring: a retrospective observational study” It is well written and important points are clearly presented. I have the following questions and comments to the authors.

1) It seems that the authors focused on MPP-CV as a primary exposure, but there were many sensitivity analyses that may distract readers. For example, MPP-VIM was equally reported as if it is a primary exposure. All sensitivity and subgroup analyses should be concisely summarized in the sensitivity and subgroup analysis section of the results. Can the authors choose a few important sensitivity and subgroup analyses?

Response: Thank for your comment. According to your suggestion, we removed MPP-VIM to the sensitivity and subgroup analyses section in the part of methods and results, and we delete the sensitivity analysis of the relationship of MPP-CV and ICU mortality.

2) Can the authors clarify how MPP was obtained by showing an equation? Some readers may not be familiar with it.

Response: Thank for your comment. We added the calculation formula of MPP in the background part (line 79-80).

3) The patient characteristics were not clear. The e-ICU data includes heterogenous patients. How many patients were elective admission? How many patients underwent cardiac and non-cardiac surgery? Can the authors adjust for them in the regression analysis?

Response: Thank for your comment. We re-extracted the type of admission and the patients who undergo cardiovascular surgery, and the proportion of the population is shown in the Table1 and supplementary table2.

Variables 

Admission type

 Medicine

 Elective surgery

 Urgent surgery

Cardiovascular surgery N (%)

3009 (49.2)

2762 (45.2)

338 (5.5)

1885 (30.8)

We then adjust for them in all the regression analyses. Specifically, age, gender, BMI, ethnicity, Charlson comorbidity index, SOFA score, admission type (elective surgery, emergency surgery or medicine), cardiovascular surgery, history of tachyarrhythmia, sepsis, incidence of AKI in the first day of ICU admission, the need for mechanical ventilation, the use of vasopressor, antihypertensive drug, sedatives, and TWA-MPP are adjusted in all the multivariable analyses.

After adding the two parameters, the main results were still robust. Except that the subgroup analysis has some difference compared with the original version. Mainly in the hypertension subgroup, after the adjustment of the two variables, the fluctuation of MPP had a more significant impact on the population with hypertension compared with those without.

4) Line 151: Was TWA-MPP adjusted? Why?

Response: Yes, TWA-MPP was adjusted. Because MPP also affects prognosis, we included TWA-MPP in the multivariable models to adjust the effect of MPP on the results.

5) Line 177: This would not be an external validation. When it comes to external validation, a newly developed prediction score is going to be tested in a different dataset. If the authors can obtain the same information from MIMIC-III, two databases can be combined into one dataset.

Response: Thank you for your comment. We corrected the expression of “external validation” to “validation test” (line 182). 

6) In this study, looking at ICU mortality would be less important.

Response: Thank for your comment. We delete the sensitivity analysis of the relationship of MPP-CV and ICU mortality.

7) Line 161: Why daytime and nighttime MPP-CV was measured? Please clarify.

Response: There is a circadian rhythm in blood pressure, and the purpose of measuring nighttime and daytime MPP-CV is to observe whether the association between MPPV and prognosis is solely contributed by daytime or nighttime MPPV (line 164-167). 

Reviewer #2: The present manuscript is an interesting study in which the authors use a large, multicenter eICU database (with external validation database) to analyze whether the coefficient of variation (CV) of mean perfusion pressure (MPP) (MPP-CV) over 24hrs within first 72hrs of ICU stay correlated with in-hospital mortality. They determined the median MPP-CV for the 2 databases. They also found that patients with high MPP-CV (especially >19%) had higher OR of in hospital mortality. They used a variety of secondary and subgroup analyses to confirm the robustness of their primary finding. They concluded that targeting fewer MPP swings in patients (~20% of average MPP) could be target for therpies. Overall, the topic is important, the manuscript is well-written, data analysis methods overall appear solid, and the findings are significant. Please see comments below for issues that should be addressed.

General comments:

1. Why were patients with CKD5 excluded? No justification is given. The authors should justify exclusions.

Response: Patients with CKD stage 5 may need to undergo dialysis, which will significantly affect MPP and increase variability. We have added relevant explanation to the method part (line112-113).

2. For exclusions, what does “extreme MPP were excluded” mean? The authors need to specify.

Response: Thank you for your comment. We specified extreme MPP in the part of data cleaning in the original manuscript. Extreme MPP refers to the values of MAP not between 0 mmHg to 150 mmHg, and the values of CVP not between -10 mmHg to 50 mmHg.

We have added relevant explanations in the exclusions in the revised version (line110-112).

3. What were variables in multivariable model and how were they chosen?

Response: Variables in multivariable model includes age, gender, BMI, ethnicity, Charlson comorbidity index, SOFA score, admission type (elective surgery, emergency surgery or medicine), cardiovascular surgery, history of tachyarrhythmia, sepsis, incidence of AKI in the first day of ICU admission, the need for mechanical ventilation, the use of vasopressor, antihypertensive drug, sedatives, and TWA-MPP.

These variables can be mainly divided into three parts. 1. Patients’ epidemiological information (age, gender, BMI, ethnicity)，2. Severity of the diseases [Charlson comorbidity index, SOFA score, admission type (elective surgery, emergency surgery or medicine), cardiovascular surgery, history of tachyarrhythmia, sepsis, incidence of AKI in the first day of ICU admission, the need for mechanical ventilation]，3.Other factors that may affect MPPV (the use of vasopressor, antihypertensive drug, sedatives, and TWA-MPP).

4. Why did the authors pick MPP-CV < or > 12% as a cutoff? Justification is not given in methods though presumably due to median of MPP-CV of 12.3% in whole dataset?

Response: Since there is no specific numerical reference from previous studies, we chose the median MPP-CV of the data set as the cut-off value (line167). 

5. Were any analyses pre-specified? This should be indicated in methods.

Response: No, it is a post hoc analysis. We clarified it in the first line of the statistical analysis part (line136).

6. Why did the authors not include hypertension or baseline blood pressure in the model? This seemingly could be a major confounder as patients with hypertension would likely have wider swings in blood pressure. They do account for this in subgroup analysis, but curious why it was not included in the original model? Is this what anti-hypertension drug refers to in the figure legend? The authors should clarify.

Response: Thank you for your suggestion and we quite agree with your point of view. However, baseline blood pressure information was not available in the database.

Considering that there may be some inaccuracy in the past history of hypertension and the high rate of missed diagnosis of hypertension, we correct the actual use of antihypertensive drugs in the multivariable models.

In addition, we adjusted for TWA-MPP in the multivariable models, which corrected for the effect of MPP on the results.

Moreover, we analyzed the hypertensive population separately in the subgroup analysis (figure 3) which showed that the fluctuation of MPP had a more significant impact on the population with hypertension compared with those without.

7. Did the authors analyze any secondary outcomes? Presumably they have information regarding outcomes such as AKI, mechanical ventilation, etc. These would be meaningful additions.

Response: Thank you for your comment. We conducted another study to analyze the relationship between MPPV and AKI which has been published in the journal of Renal Failure (Peng Y, Wu B, Xing C, Mao H. Increased mean perfusion pressure variability is associated with subsequent deterioration of renal function in critically ill patients with central venous pressure monitoring: a retrospective observational study. Ren Fail. 2022;44(1):1976-1984.). The result showed that increased MPPV was associated with an increased risk of subsequent deterioration of renal function in critically ill patients with central venous pressure monitoring. Maintaining stable MPP may reduce the risk of renal function deterioration.

Reviewer #3: The authors submit a manuscript describing a retrospective analysis of large database of critically ill patients having invasive monitoring. The focus was on mean-perfusion pressure variation (MAP-CVP) and mortality. The found that the average coefficient of variation the first 24 hrs of ICU admission was higher in non-survivors versus survivors. The highest decile of MPP CV was independently associated with mortality.

The topic of this paper is clinically interesting and consistent with other reports. I have a few comments.

1. It would be helpful for the readers to have an understanding of the unique aspects of the question of this study. That is, what new questions the study asks. They mention that there is knowledge that MPP variation is associated with adverse outcomes then how does this paper add to this body of knowledge? Interestingly, there are other data not references showing that fluctuations in BP in patients undergoing surgery is associated with adverse events (I believe the first author was Aronson, S). They did not address MPP to my knowledge.

Response: Thank you for your comment. Previous studies have shown that MPP is associated with poor prognosis, but no studies have shown a correlation between MPP variability and prognosis. Variability is a characteristic of MPP which is not equal to MPP and is independent of MPP itself. 

Before writing the article, we also searched PubMed for the relationship between blood pressure variability and prognosis. The variability of other blood pressure indicators is also related to prognosis, including MAPV and SBPV. However, there are many studies in this area which mainly focused on patients with hypertension or patients receiving surgery. Our article focused on the MPPV of critically ill patients in the ICU, so the article [1] on the relationship between SBPV and prognosis during surgery was not cited before. In the revised version, we cite this article in the discussion section (line292-293). 

[1] Aronson S, Stafford-Smith M, Phillips-Bute B, et al. Intraoperative systolic blood pressure variability predicts 30-day mortality in aortocoronary bypass surgery patients. Anesthesiology. 2010;113(2):305-312.

2. Page 5 in the Methods section line 112: Readers may want to know how you defined extreme MPP data.

Response: Thank you for your comment. We specified extreme MPP in the part of data cleaning in the original manuscript. Extreme MPP refers to the values of MAP not between 0 mmHg to 150 mmHg, and the values of CVP not between -10 mmHg to 50 mmHg.

We have added relevant explanations in the exclusions in the revised version (line110-112).

3. Page 6, lines 164-165: how did you decide to categorize high and low variability as MPP 12%?

Response: Since there is no specific numerical reference from previous studies, we chose the median MPP-CV of the data set as the cut-off value (line167). 

4. An entrance criteria of having both direct blood pressure and CVP present may limit the external validity of the study when those measures are not present. That is why the readers need to know the reason for exclusion from analysis. One wonders what the added value of MPP versus MAP alone might be in the predictive modeling. While I agree with the rationale for MPP is there useful data on MAP CV alone? We do not know if the relationship found in this study is due to reduced perfusion per se versus just MAP or the factors that cause MAP variation. How do we know as an example this is not simply due to HFpEF? Those patients have high variability due to diastolic dysfunction.

Response: Thank you for your comment. 

(1) Inclusion of patients with CVP monitoring may make the results not extrapolated well. We showed the relationship between MAP-CV and prognosis in Supplementary Figure 2. In the two databases, the results showed no significant difference, suggesting that MPP-CV may be more robust in predicting hospital mortality. 

Supplementary Fig.2 The associations between in-hospital mortality risk and MAP-CV in both databases.

Furthermore, in terms of predicting hospital mortality, MPPV has a slightly advantage than MAPV, which has been shown in supplementary table3.

Supplementary Table 3. The comparison of the AUC between MPPV and MAPV in prediction the hospital mortality.

 AUC of MPPV (95% CI) AUC of MAPV (95% CI)

CV 0.56 (0.54-0.58) 0.50 (0.48-0.52)

VIM 0.54 (0.52-0.56) 0.50 (0.48-0.52)

In addition, correlation analysis (Supplementary Fig.1) showed that the correlation coefficient between MPP-CV and MAP-CV was 0.77, r ²= 0.60 which indicated that approximately 60% of MPP-CV can be explained by MAP-CV. Therefore, although MPP-CV is mostly determined by MAP-CV, they are not completely equivalent.

(2) We also briefly analyzed the relationship between CVP-CV and hospital mortality. The result is shown in the following figure

After adjustment, the correlation between CVP variability (CVP-CV) and in-hospital mortality was not obvious.

(3) We have included TWA-MPP in the multivariable models to adjust the effect of MPP on the results. So, we consider that the results we got were due to the variability of MPP rather than the decrease in MPP. 

(4) Admittedly, HFpEF can affect blood pressure variability. But this research population is focused on critically ill patients, and is hard to obtain echocardiographic data. Based on the database, it is difficult to answer your question satisfactorily, and further research may be needed in the future.

5. Lines 292-293. MAP is the product of cardiac output and SVR. MPP is MAP -CVP.

Response: Thank you for your comment. Actually, the formula in the references [2] we quoted is indeed shown in the manuscript. In the original text, it was written as “SVR = ([(MAP–CVP]/CO) × 80)”. The detailed relationships between pressure-output-resistance are clarified in this review. [3].

[2] Chotalia M, Ali M, Hebballi R, Singh H, Parekh D, Bangash MN, et al. Hyperdynamic Left Ventricular Ejection Fraction in ICU Patients With Sepsis. Crit Care Med. 2021. 32. Meng L: 

[3] Meng L, Heterogeneous impact of hypotension on organ perfusion and outcomes: a narrative review. Br J Anaesth 2021, 127(6):845-861.

6. Lines 350-351: Making as statement that MPP should be targeted as a means to improve outcome is not really warranted based on these data.

Response: Thank you for your comment. Yes, the exact value of MPP-CV may not be a target for improving prognosis, but what we meant in the original manuscript is that keeping MPP-CV < 20% as far as possible may be a target for improving prognosis, that is, severe MPP fluctuations may need to be avoided in the management of critically ill patients. We have changed our expression in the revised version (line356-359).

---

## [Decision Letter · Decision Letter 1]

29 May 2023

Severe fluctuation in mean perfusion pressure is associated with increased risk of in-hospital mortality in critically ill patients with central venous pressure monitoring: a retrospective observational study

PONE-D-23-05624R1

Dear Dr. Mao,

We’re pleased to inform you that your manuscript has been judged scientifically suitable for publication and will be formally accepted for publication once it meets all outstanding technical requirements.

Kind regards,

Karthik Raghunathan

Academic Editor

PLOS ONE

Additional Editor Comments (optional):

Congratulations!

Reviewers' comments:

Reviewer's Responses to Questions

**Comments to the Author**

1. If the authors have adequately addressed your comments raised in a previous round of review and you feel that this manuscript is now acceptable for publication, you may indicate that here to bypass the “Comments to the Author” section, enter your conflict of interest statement in the “Confidential to Editor” section, and submit your "Accept" recommendation.

Reviewer #1: All comments have been addressed

Reviewer #2: All comments have been addressed

2. Is the manuscript technically sound, and do the data support the conclusions?

Reviewer #1: Yes

Reviewer #2: Yes

3. Has the statistical analysis been performed appropriately and rigorously? 

Reviewer #1: Yes

Reviewer #2: Yes

4. Have the authors made all data underlying the findings in their manuscript fully available?

Reviewer #1: Yes

Reviewer #2: Yes

5. Is the manuscript presented in an intelligible fashion and written in standard English?

Reviewer #1: Yes

Reviewer #2: Yes

6. Review Comments to the Author

Reviewer #1: (No Response)

Reviewer #2: The authors have adjusted language and supplied additional data. Authors have satisfactorily responded to my critiques.

7. PLOS authors have the option to publish the peer review history of their article (what does this mean?). If published, this will include your full peer review and any attached files.

Reviewer #1: No

Reviewer #2: No

---

## [Editor Report · Acceptance letter]

5 Jun 2023

PONE-D-23-05624R1 

Severe fluctuation in mean perfusion pressure is associated with increased risk of in-hospital mortality in critically ill patients with central venous pressure monitoring: a retrospective observational study 

Dear Dr. Mao:

I'm pleased to inform you that your manuscript has been deemed suitable for publication in PLOS ONE. Congratulations! Your manuscript is now with our production department. 

Kind regards, 

on behalf of

Dr. Karthik Raghunathan 

Academic Editor

PLOS ONE